The comprehensive analysis of the prognostic and functional role of N-terminal methyltransferases 1 in pan-cancer

Tan Lifan 1
Li Wensong 1
Su Qin 2 18980500361@163.com
1 Department of Otolaryngology, West China-Guang’an Hospital, Sichuan University , Guang’an, Sichuan , China
2 Department of Otolaryngology, The People’s Hospital of Dujiangyan , Dujiangyan, Sichuan , China
Uversky Vladimir
Electronic publication date: 2023 Oct 24
Publication date: 2023
Volume: 11
Electronic Location ID: e16263
Received 2023 Jul 24; Accepted 2023 Sep 18
Copyright: © 2023 Tan et al.
Copyright year: 2023
Copyright holder: Tan et al.
License: This is an open access article distributed under the terms of the Creative Commons Attribution License, which permits unrestricted use, distribution, reproduction and adaptation in any medium and for any purpose provided that it is properly attributed. For attribution, the original author(s), title, publication source (PeerJ) and either DOI or URL of the article must be cited.
License URL: https://creativecommons.org/licenses/by/4.0/

Keywords: NTMT1, Pan-cancer, Bioinformatics analysis, Prognostic, Head and neck squamous cell carcinoma

Funding: The authors received no funding for this work.

==============================
Background

NTMT1, a transfer methylase that adds methyl groups to the N-terminus of proteins, has been identified as a critical player in tumor development and progression. However, its precise function in pan-cancer is still unclear. To gain a more comprehensive understanding of its role in cancer, we performed a thorough bioinformatics analysis.

Methods

To conduct our analysis, we gathered data from multiple sources, including RNA sequencing and clinical data from the TCGA database, protein expression data from the UALCAN and HPA databases, and single-cell expression data from the CancerSEA database. Additionally, we utilized TISIDB to investigate the interaction between the tumor and the immune system. To assess the impact of NTMT1 on the proliferation of SNU1076 cells, we performed a CCK8 assay. We also employed cellular immunofluorescence to detect DNA damage and used flow cytometry to measure tumor cell apoptosis.

Results

Our analysis revealed that NTMT1 was significantly overexpressed in various types of tumors and that high levels of NTMT1 were associated with poor survival outcomes. Functional enrichment analysis indicated that NTMT1 may contribute to tumor development and progression by regulating pathways involved in cell proliferation and immune response. In addition, we found that knockdown of NTMT1 expression led to reduced cell proliferation, increased DNA damage, and enhanced apoptosis in HNSCC cells.

Conclusion

High expression of NTMT1 in tumors is associated with poor prognosis. The underlying regulatory mechanism of NTMT1 in cancer is complex, and it may be involved in both the promotion of tumor development and the inhibition of the tumor immune microenvironment.

Introduction

In recent years, there has been increasing interest in studying the role of epigenetic modifications in tumor invasion and metastasis (Huo & Zhang, 2021; Yankova & Blackaby, 2021; Zhang & Song, 2021). One of the most common modifications found in mRNA is N6-methyladenine (m6a), which regulates the expression of oncogenes and tumor suppressor genes through methylation transferase proteins. This process can promote tumor occurrence, invasion, and progression (Wu & Sang, 2018). Research has shown that several members of the METTL family play key roles in cancer development. For example, METTL3 has been found to have an oncogenic role in leukemia (Vu & Pickering, 2017), glioma (Chai & Chang, 2021; Chang & Chai, 2021), and hepatocellular carcinoma (Pan & Lin, 2021). METTL7B has also been shown to promote thyroid cancer cell proliferation and metastasis by inducing EMT (Ye & Jiang, 2019). Furthermore, METTL7B is overexpressed in lung cancer, where it promotes tumor occurrence and progression (Liu & Li, 2020).

NTMT1 (N-terminal methyltransferases 1), also known as METTL11A or NRMT1, is the first identified eukaryotic N-terminal methyltransferase (Tooley & Petkowski, 2010). This enzyme is responsible for adding a methyl group to the N-terminal of specific proteins, which is crucial for their localization and function within the cell. NTMT1 has been found to have a variety of functions in different cellular processes, including regulation of mitosis, DNA damage repair, neurogenesis, and stem cell maintenance (Tooley & Petkowski, 2010; Varshavsky, 2019; Cai & Fu, 2014). For instance, NTMT1 knockdown significantly enhances the sensitivity of breast cancer cell lines to both etoposide treatment and γ-irradiation, increases proliferation rate, invasive potential, anchorage-independent growth, as well as, xenograft tumor size, and tamoxifen sensitivity (Bonsignore & Butler, 2015; Dong & Mao, 2015). Meanwhile, it has been suggested that NTMT1 promotes cervical cancer cell migration by upregulating ELK3 (Zhang & Song, 2021). This suggests that NTMT1 may act as a tumor suppressor in some contexts. However, its role in cancer development and progression appears to be complex and context-dependent. In other types of cancer, such as cervical cancer and colorectal cancer, NTMT1 has been found to act as an oncogene, promoting the migration and growth of cancer cells (Zhang & Song, 2021; Shields & Tooley, 2017). In cervical cancer, high levels of NTMT1 expression have been associated with poorer prognosis and increased tumor aggressiveness (Zhang & Song, 2021). The exact mechanisms by which NTMT1 promotes or suppresses cancer development and progression are not yet fully understood. However, it is thought to be related to the regulation of various signaling pathways involved in cell growth and survival.

Researchers are actively exploring the potential of developing NTMT1-targeted therapies for cancer treatment. One promising approach is to develop small molecule inhibitors that can selectively target NTMT1 activity (Deng & Dong, 2023; Dong & Iyamu, 2022; Mackie & Chen, 2020; Dong & Deng, 2022). Recent studies have shown that a degrader 1 targeting NTMT1 has significant anti-proliferative activity on HCT116 cells in both 2D and 3D culture environments by inducing cell cycle arrest at the G0/G1 phase (Zhou & Wu, 2022). Furthermore, label-free global proteomic analysis revealed that degrader 1 induced overexpression of calreticulin (CALR), an immunogenic cell death (ICD) signaling protein known to elicit anti-tumor immune responses and clinically associated with high survival in colorectal cancer patients (Zhou & Wu, 2022). These findings provide a valuable avenue to investigate the biological functions of NTMT1 in cancer and to develop new therapeutic strategies.

Taken together, NTMT1 plays a complex and context-dependent role in cancer initiation and progression. Further studies are needed to fully understand the mechanism of action in different types of cancer. In this study, the expression of NTMT1 in a variety of tumors was investigated using bioinformatics techniques, with particular emphasis on its role in the development and progression of head and neck squamous cell carcinoma (HNSCC) and its relationship with immunity. These results provide new insights into the potential treatment of a variety of tumors.

Materials and Methods

Data collection

To evaluate NTMT1 expression in cancer patients, we obtained mRNA expression profiles and clinical outcome data from The Cancer Genome Atlas (TCGA) pan-cancer cohort website (https://portal.gdc.cancer.gov/). Additionally, mRNA expression profiles for normal tissues were obtained from both the TCGA (https://portal.gdc.cancer.gov/) and the University of California Santa Cruz (UCSC) Xena (https://xenabrowser.net/datapages/). To ensure consistency and comparability between samples, we standardized the raw data by normalizing via the transcripts per million (TPM) method. We then applied a log2(TPM+1) transformation for further analysis.

Expression analysis of NTMT1

We conducted a comparative analysis of NTMT1 mRNA expression levels in tumors and corresponding normal tissues using mRNA expression profiles. This analysis included 33 different types of tumors, and we also compared NTMT1 expression between patients with different characteristics. We analyzed the expression levels of NTMT1 protein in tumors using Human Protein Atlas (HPA) database (http://www.proteinatlas.org) (Colwill & Renewable Protein Binder Working, 2011) and UCLCAN databases (https://ualcan.path.uab.edu/index.html) (Chandrashekar & Karthikeyan, 2022). To further investigate NTMT1 expression, we retrieved representative immunohistochemistry and immunofluorescence images from the HPA (Colwill & Renewable Protein Binder Working, 2011).

Survival analysis

We used GEPIA2.0 to investigate the potential prognostic significance of NTMT1 expression in various cancer types by applying the Mantel–Cox test to evaluate overall survival (OS) and disease-free survival (RFS) (Tang & Kang, 2019). To perform a more comprehensive survival analysis, we divided TCGA tumor patients into low and high NTMT1 expression groups using a 50% cut-off value, and then conducted proportional hazards hypothesis testing and fitted survival regression models using the survival package (3.3.1). Finally, we used the survminer package (3.3.6) and ggplot2 package (3.3.6) to visualize the results (Liu & Lichtenberg, 2018). Additionally, we used the Kaplan–Meier plotter (http://kmplot.com/analysis) to analyze the relationship between NTMT1 expression and the survival of tumor patients by applying the log-rank test (Nagy & Munkacsy, 2021).

Immune infiltration analysis of NTMT1

TISIDB (http://cis.hku.hk/TISIDB/index.php) is a comprehensive web portal that enables analysis of the interactions between tumors and the immune system by integrating various types of heterogeneous data (Ru & Wong, 2019). In our study, we used TISIDB to explore the associations between the expression of NTMT1 and the abundance of tumor-infiltrating lymphocytes (TILs), immunomodulators, and chemokines. Moreover, we utilized the ssGSEA algorithm, which is available in the R package-GSVA (Hanzelmann & Castelo, 2013) to evaluate immune cell infiltration in HNSCC. To conduct these analyses, we utilized a panel of 24 different immune cell types (Bindea & Mlecnik, 2013), which included dendritic cells (DCs), CD8+ T cells, natural killer (NK) cells, mast cells, type 17 Th (Th17) cells, regulatory T cells (Treg), type 1 Th (Th1) cells, macrophages, among others. Furthermore, we used the estimate R package (1.0.13) (Yoshihara & Shahmoradgoli, 2013) to calculate both the StromalScore and ImmuneScore for HNSCC, providing valuable insights into the potential roles of the tumor microenvironment in HNSCC progression and treatment.

Genetic alteration analysis

We used the cBioPortal (Gao & Aksoy, 2013) (https://www.cbioportal.org/) to gather data on the frequency of alterations and mutation types of NTMT1 in all TCGA tumors. Furthermore, we downloaded and analyzed OS data for TCGA cancer patients with and without NTMT1 gene alterations using the “Comparison” module. In the “Comparison” module, we analyzed the relationship between NTMT1 gene alterations and OS of TCGA tumor patients.

Single-cell sequencing

CancerSEA (http://biocc.hrbmu.edu.cn/CancerSEA/home.jsp) is the first dedicated database that aims to comprehensively decode distinct functional states of cancer cells at single-cell resolution (Yuan & Yan, 2019). In our study, we utilized this database to analyze the correlation between NTMT1 expression and various tumor functions based on single-cell sequencing data. The T-SNE diagrams were used to demonstrate the expression profiles of NTMT1 at the single-cell level in TCGA samples.

Protein-protein interaction network analysis and functional enrichment analysis

We utilized the BioGRID4.4 website (https://thebiogrid.org/) to analyze the protein-protein interaction network associated with NTMT1 (Oughtred & Rust, 2021). The top 50 NTMT1-associated proteins were obtained via the STRING database (https://cn.string-db.org/), with a medium confidence cut-off of 0.4 for the minimum required interaction score. Additionally, we used GEPIA2 (Tang & Kang, 2019) (http://gepia2.cancer-pku.cn/#index) to obtain the top 100 NTMT1-correlated genes from TCGA tumor samples. We then utilized UALCAN to identify NTMT1-associated genes in HNSCC and conducted an intersecting analysis with the top 100 genes to identify HNSCC-related genes.

To identify differentially expressed genes (DEGs) between high- and low-NTMT1 expression groups, we utilized the “DESeq2” (v1.36.0) R package (Love & Huber, 2014) with |logFC| > 1 and p < 0.05 as the threshold. The volcano plots of the DEGs were visualized using the “ggplot2” (v3.3.6) R package. The “org.Hs.eg.db” R package was used to convert the Entrez ID to the gene symbol. The “ClusterProfiler” (Yu & Wang, 2012) (v4.4.4) R package was used for the functional annotation and Gene Set Enrichment Analysis (GSEA) of the DEGs between high- and low-NTMT1 expression groups. To perform GSEA, we selected the curated reference gene sets from the MgDB file: c2.cp.all.v2022.1.Hs.symbols.gmt (https://www.gsea-msigdb.org/gsea/msigdb/index.jsp) (Subramanian & Tamayo, 2005).

Promoter methylation of NTMT1 in pan-cancer

UALCAN (Chandrashekar & Karthikeyan, 2022) (http://ualcan.path.uab.edu) is a web portal that utilizes RNA sequencing and clinical data from the TCGA database across 31 different types of cancer. In our study, we utilized UALCAN to investigate the differential methylation levels of NTMT1 in various types of tumors.

Cell culture and regents

SNU1076 cell lines were purchased from the Cell Bank of the Shanghai Institute of Cells, Chinese Academy of Science (Shanghai, China). The cell was cultured in RPMI 1640 medium (Gibco, Billings, MT, USA) containing 10% fetal bovine serum (FBS, BI), 100 U/mL penicillin, and 100 μg/Ml streptomycin at 37 °C in a 5% CO2 environment. RiboPECTTM CP Transfection Kit was obtained from Ribobio (Guangzhou, China). RNA Quick Purification kit was purchased from ESscience. PrimeScript RT Master Mix was purchased from TAKARA. CCK8 Kit was purchased from Biosharp. SiRNA NTMT1 was synthesized from Ribobio (Guangzhou) were synthesized using sequences provided in the relevant literature (Zhang & Song, 2021). siNTMT1-1: GCAUUGGGAGGAUCACCAATT, siNTMT1-2: GCAAGAGGGTGAGGAAC TA. Anti-γ-H2AX was purchased from CST. Annexin V-FITC/PI apoptosis kit was purchased from MultiSciences.

RNA extraction and qPCR

Cell transfection was performed using the riboPECTTM CP Transfection Kit according to the manufacturer’s protocol. Total RNA from approximately 1 × 106 cells was isolated using an RNA Quick Purification kit. The primers used for qPCR, including those for NTMT1 and GAPDH, were synthesized from Shenggong (Shanghai, China). The primer sequences were (5′–3′): NTMT1 forward-CGAGGTGATAGAAGACGAGAAGC, reverse-CGGGAGCTGTTGATGTCGAT. GAPDH forward-GGAGCGAGATCCCTCCAAAAT, reverse-GGCTGTTGTCATACTTCTCATGG. Relative elevels were normalized to GAPDH and calculated according to the 2−ΔΔCT method.

Cell proliferation

A total of 5 × 103 cells were seeded on 96-well plates in a culture medium containing 10% FBS with four replicate wells for each group. Transfection was performed the next day. At various time points, cell viability was measured using Counting kit-8 according to the manufacturer’s instructions.

Cellular immunofluorescence

The cells were fixed with 4% paraformaldehyde for 30 min at room temperature and treated with 0.1% Triton-X for 10 min. Next, they were blocked with 5% BSA for 1 h at room temperature and incubated with anti-γ-H2AX (1:500) (CST) overnight at 4 °C. This was followed by incubation with CoraLite594-conjugated Goat Anti-Rabbit IgG (H+L) (1:500; Proteintech, Rosemont, IL, USA) at room temperature for 1 h in the dark. Nuclei were stained with DAPI in the dark for 5 min, and the fluorescence intensity was observed under a microscope.

Detection of apoptosis

To determine apoptosis, cells were seeded at a density of 1 × 105 in 12-well plates. The following day, the cells were treated with siRNA for 72 h. After collecting the cells and washing them with 1× PBS, the pellet was gently resuspended in 500 µL of 1X binding buffer. Annexin V-FITC (5 µL) and PI (10 μL) were added to the cell suspension, which was incubated for 5 min in the dark. At the same time, the single dye tube of FITC and PI were set up. The samples were detected by Beckman CytoFLEX, and the analysis was performed using FlowJo v. 10.8.1 software from the USA.

Statistical analysis

The comparison between the two groups was performed using the Wilcoxon rank-sum test, while the correlation between the two groups was evaluated using Spearman’s correlation coefficient. Survival analysis was conducted using the Kaplan–Meier method and log-rank test. Statistical analysis was conducted using R (version 4.2.1) and GraphPad Prism 9 software (USA), ns, no significant, *p < 0.05, **p < 0.01, and ***p < 0.001.

Results

NTMT1 mRNA and protein expression levels in pan-cancer

In this study, the expression of NTMT1 was analyzed in 33 cancer datasets obtained from the TCGA database. The findings demonstrated that NTMT1 was significantly up-regulated in bladder urothelial carcinoma (BLCA), breast invasive carcinoma (BRCA), cholangiocarcinoma (CHOL), colon adenocarcinoma (COAD), esophageal carcinoma (ESCA), head and neck squamous cell carcinoma (HNSCC), liver hepatocellular carcinoma (LIHC), lung adenocarcinoma (LUAD), lung squamous cell carcinoma (LUSC), prostate adenocarcinoma (PRAD), rectum adenocarcinoma (READ), stomach adenocarcinoma (STAD), thyroid carcinoma (THCA), and uterine corpus endometrial carcinoma (UCEC). Conversely, NTMT1 was significantly down-regulated in kidney renal clear cell carcinoma (KIRC) and pheochromocytoma and paraganglioma (PCPG) (Fig. 1A). The analysis of uniform TCGA_GTEx data collected from UCSC also revealed a significant up-regulation of NTMT1 in various types of cancer tissues, which was largely consistent with the results obtained from TCGA (Fig. 1B). Moreover, the expression of NTMT1 in 18 types of tumors with paired samples in TCGA was examined, and NTMT1 was found to be significantly up-regulated in 12 types of cancer tissues (Fig. 1C).

Figure 1 NTMT1 mRNA expression in various tumors.

(A) The mRNA expression of NTMT1 in pan-cancer in TCGA database. (B) NTMT1 expression differences between tumor and normal tissues in pan-cancer from the TCGA_GTEx database. (C) The mRNA expression of NTMT1 in cancer and para-cancer paired samples in TCGA. ns, p > 0.05; *p < 0.05; **p < 0.01; ***p < 0.001.

We further investigated the protein expression levels of NTMT1 in both normal and tumor tissues of various human organs using the HPA. Our findings revealed high protein expression levels of NTMT1 in BRCA, COCA, LUAD, HNSCC, LUSC, and PAAD (Figs. 2A and S1A). The protein expression levels were statistically different between different subtypes (Figs. S1B and S1C) (Zhang & Chen, 2022; Chen & Chandrashekar, 2019). We also extracted representative immunohistochemistry (IHC) images of the tumor and normal tissues from HNSCC (Fig. 2B) and BRCA (Fig. 2C), suggesting that NTMT1 may play a critical role in the development and progression of these types of cancer. Furthermore, subcellular localization analysis using immunofluorescence demonstrated that NTMT1 is primarily located in the nucleus and cytoplasm of A-431 and U2OS cells (Fig. 2D). These results suggest that NTMT1 might play a crucial role in the occurrence and progression of tumors. However, further research is required to gain a better understanding of the functions of NTMT1 in cancer development and to assess the potential of NTMT1-targeted therapies for cancer treatment.

Figure 2 NTMT1 protein expression in various tumors.

(A) The expression level of NTMT1 protein between normal tissues and primary tissue of BRCA, HNSCC, COCA, and LUAD based on the CPTAC dataset, ***p < 0.001. (B and C) The IHC images of NTMT1 in normal and tumor tissues of HNSCC (B) and BRCA (C) extracted from the HPA. (D) Immunocytochemistry for determining the subcellular location of NTMT1 in A431 and U2OS cell lines by HPA. NTMT1 localized to the nucleoplasm, cytosol (green). Microtubules were stained in red and the nucleus was stained in blue (DAPI). The antibody of NTMT1 was HPA020092.

The association between NTMT1 expression and prognosis in pan-cancer

To evaluate the prognostic significance of NTMT1 in pan-cancer, we conducted a Kaplan–Meier survival analysis to investigate the correlation between NTMT1 expression and clinical outcomes. Initially, we examined the relationship between NTMT1 expression and OS in 33 different types of cancer. Our findings revealed that high expression of NTMT1 was positively associated with poor OS in HNSCC, ACC, and LAML (Figs. 3A and 3B). Moreover, high expression of NTMT1 was positively correlated with poor DSS and recurrence-free survival (RFS) in HNSCC, ACC, and READ (Figs. 3C and 3D), as well as with poor progression-free interval (PFI) in HNSCC, ACC, and UVM (Fig. 3E). Furthermore, the results of Kaplan–Meier plotter analysis indicated that high NTMT1 expression was positively associated with worse OS in BLCA, HNSCC, and STAD (Fig. S2A), and worse RFS in LIHC, LUSC, and STAD (Fig. S2B).

Figure 3 Prognostic values of NTMT1 expression in pan-cancer.

(A) The effects of NTMT1 gene expression on OS of pan-cancer, including OS (overall survival) by GEPIA2.0. (B) The effects of NTMT1 expression on OS in ACC, HNSCC, and LAML, respectively. (C) The effects of NTMT1 gene expression on DFS of pan-cancer, including OS by GEPIA2.0. (D) Survival curves of NTMT1 on DSS (disease-specific survival) in ACC, HNSCC, and READ, respectively. (E) Survival curves of NTMT1 on PFI (progress-free interval) in ACC, HNSCC, and READ, respectively.

These results suggest that NTMT1 may have a critical role in the development and progression of these specific types of cancer. Furthermore, NTMT1 expression could serve as a valuable prognostic biomarker in these cancer types, which could help improve patient outcomes through earlier detection and personalized treatment approaches. However, further research is required to better comprehend the molecular mechanisms underlying the role of NTMT1 in these cancers and to evaluate the potential of NTMT1-targeted therapies for cancer treatment.

Clinical relevance of NTMT1 expression

To assess the diagnostic value of NTMT1 in pan-cancer, we performed a receiver operating characteristic (ROC) curve analysis based on data from the Cancer Genome Atlas (TCGA) database. Our analysis revealed that NTMT1 exhibited a high degree of diagnostic accuracy for eight types of cancer (AUC > 0.9), including PCPG, CHOL, COAD, READ, DLBC, LAML, TGCT, and THYM (Fig. 4A). These findings suggest that NTMT1 expression levels may be useful in predicting the clinical outcome and may help clinicians tailor treatment plans for these patients. However, the AUC value of 0.878 for HNSCC suggests that NTMT1 expression has only a moderately accurate diagnostic value for HNSCC compared to the other types of cancer. Nevertheless, our previous analysis showed a positive correlation between high NTMT1 expression and poor overall survival in HNSCC patients, indicating that NTMT1 may have prognostic value in this cancer type. Further research is needed to fully understand the clinical significance of NTMT1 in HNSCC and to identify potential therapeutic strategies targeting NTMT1 in this cancer type.

Figure 4 The correlation between NTMT1 gene expression and clinical characteristics.

(A) Validation of diagnostic value of NTMT1 for PCPG, CHOL, COAD, READ, DLBC, LAML, TGCT, and THYM using ROC curve. (B) The expression levels of the NTMT1 gene were analyzed by the main clinical features (pathological N stages, T stage, pathologic stage, histologic grade, primary therapy outcome, lymphovascular invasion, gender, and lymphnode neck dissection) of HNSCC. Log2 (TPM+1) was applied for log scale. ns, p > 0.05; *p < 0.05; **p < 0.01; ***p < 0.001.

Given the significant prognostic value of NTMT1 in HNSCC, our study aimed to investigate the relationship between NTMT1 expression and clinicopathological features in HNSCC. Our results showed that NTMT1 expression was significantly associated with several clinical parameters, including pathological N stages, T stage, pathologic stage, histologic grade, primary therapy outcome, lymphovascular invasion, gender, and lymph node neck dissection (Fig. 4B). These findings suggest that NTMT1 may play a critical role in the development and progression of HNSCC and that NTMT1 expression could serve as a potential biomarker for the diagnosis and prognosis of HNSCC. However, further research is needed to validate the clinical utility of NTMT1 as a diagnostic and prognostic indicator in HNSCC and to explore the molecular mechanisms underlying the role of NTMT1 in HNSCC.

NTMT1 mutation in various tumors

Using TCGA data, we analyzed the mutation status of the NTMT1 gene through the cBioPortal platform to explore gene mutation patterns in various tumors. Pan-cancer analysis revealed NTMT1 high amplification in ESCA, PAAD, BRCA, OV, and SARC. The rate of “deep deletion” in BRCA, OV, and melanoma was the highest, while mutations mainly occur in ESCA and OV (Fig. 5A). As shown in Fig. 5B, the overall mutation frequency of NTMT1 in pan-cancer is only 2.4%. We also observed a positive correlation between NTMT1 mRNA expression and copy number (Spearman: 0.56; Fig. 5C). Finally, we investigated the potential association between NTMT1 gene alterations and overall survival in pan-cancer patients, and found that patients with NTMT1 gene alterations had significantly lower OS compared to those without NTMT1 gene alterations (p = 0.0118; Fig. 5D), as indicated by our analysis using Kaplan–Meier survival curves and log-rank tests.

Figure 5 NTMT1 gene mutation in various cancers.

(A) Alteration frequency with the mutation type of NTMT1 in human pan-cancer. (B) OncoPrint of type and frequency of NTMT1 gene mutations in pan-cancer TCGA sample. (C) The relationship between copy number alteration and mRNA expression of NTMT1. (D) Kaplan–Meier survival curves show the OS of pan-cancer patients with or without NTMT1 gene alterations.

Promoter methylation of NTMT1 in human cancers

Research has shown that DNA methylation plays a critical role in the development and pathogenesis of various diseases, and is often associated with transcriptional repression. These findings highlight the importance of epigenetic modifications, such as DNA methylation, in regulating gene expression and contributing to disease pathophysiology (Smith & Sen, 2020). In this study, we investigated the methylation status of the NTMT1 gene in normal and tumor tissues by comparing the methylation levels of the NTMT1 promoter region (Figs. 6 and S3). Our analysis revealed that the methylation level of the NTMT1 promoter region was significantly higher in several tumor tissues, including BRCA, CHOL, COAD, KIRP, LUAD, PAAD, READ, SARC, and THCA (Fig. S3). Conversely, the level of NTMT1 methylation was significantly lower in several tumor tissues, including BLCA, CESC, HNSCC, KIRC, LIHC, LUSC, PRAD, TGCT, and UCEC (Fig. 6). Previous studies have suggested that decreased promoter region methylation can lead to increased gene expression, which in turn could contribute to the development and progression of certain types of cancer. Therefore, the reduced NTMT1 promoter region methylation observed in these tumor tissues, including BLCA, CESC, HNSCC, KIRC, LIHC, LUSC, PRAD, TGCT, and UCEC, may suggest a potential role for NTMT1 in the pathogenesis of these cancers. However, there was no significant difference in the promoter region methylation level of NTMT1 in ESCA, GBM, PCPG, STAD, and THYM. These findings suggest that the methylation status of the NTMT1 promoter region may play a role in tumorigenesis. However, further research is needed to fully understand the functional significance of NTMT1 and its relationship with promoter region methylation in these tumors.

Figure 6 Promoter methylation levels of NTMT1 in cancers.

The methylation level of the NTMT1 promoter region in BLCA, CESC, HNSCC, KIRC, LIHC, LUSC, PRAD, TGCT, and UCEC from UALCAN.

The potential mechanisms by which NTMT1 could contribute to the development and progression of these cancers are not yet fully understood and require further investigation. However, previous studies have suggested that NTMT1 may play a role in modulating various cellular processes, including mRNA splicing, translation, and protein stability (Bade & Cai, 2021). Moreover, recent studies have shown that NTMT1 can also modulate the activity and stability of several tumor suppressor proteins, including p53 (Sathyan & Fachinetti, 2017), which are commonly mutated or silenced in various types of cancer.

The relationship between NTMT1 expression and the tumor immune microenvironment

We performed a co-expression analysis on NTMT1 and immune-related genes (Ru & Wong, 2019) associated with immunoinhibitors, immunostimulators, MHC molecules, tumor-infiltrating lymphocytes, chemokines, and chemokine receptors in pan-cancer (Fig. 5). Our analysis revealed that, in the majority of cancers, there were negative correlations between NTMT1 and chemokine receptors (Figs. 7A–7F). However, in BLCA, PCPG, SARC, and LGG, the majority of immune-related genes displayed a positive correlation with NTMT1, suggesting that NTMT1 expression may be a useful biomarker for identifying patients who could benefit from immunotherapy. Additionally, targeting NTMT1 may enhance the effectiveness of immunotherapies by modulating the immune response in these cancers. Conversely, we observed negative correlations between NTMT1 and immune-related genes in HNSCC, KICH, and COAD, suggesting that immunotherapy may not be as effective in these cancers. However, further research is needed to fully understand the clinical implications of these findings and to determine the potential utility of NTMT1 as a therapeutic target in these cancers. However, further research is needed to fully understand the clinical implications and to determine the potential utility of NTMT1 as a therapeutic target in these cancers.

Figure 7 Correlation of NTMT1 and immunoregulation‑related genes in pan‑cancer.

Correlations between NTMT1 expression and (A) tumor‑infiltrating lymphocytes (TILs), (B) chemokines receptors, (C) chemokines, (D) MHC molecules, (E) im-munostimulators, (F) immunoinhibitors.

The immune microenvironment plays a crucial role in the development and progression of tumors. Given the crucial role of NTMT1 expression in HNSCC, we investigate the relationship between NTMT1 and the immune microenvironment in HNSCC, we utilized the ssGSEA algorithm to examine the correlation between NTMT1 expression and immune cells in HNSCC. Our analysis revealed that elevated NTMT1 expression was associated with reduced infiltration of aDC, iDC, cytotoxic cells, CD8 T cells, B cells, mast cells, and T cells (Fig. S4A). Furthermore, the high NTMT1 expression group displayed lower stromal score, immune score, and ESTIMATE score (Fig. S4B). We also observed reduced expression of multiple immune markers in the high NTMT1 expression group (Figs. S4C–S4G). The reduced expression of multiple immune markers in the high NTMT1 expression group could indicate a weakened immune response against cancer cells, suggesting that NTMT1 may be involved in the evasion of immune surveillance and thus promoting the progression and metastasis of HNSCC. This finding highlights the potential of NTMT1 as a therapeutic target for enhancing the immune response and improving the clinical outcomes of HNSCC patients. These findings suggest that NTMT1 may play a role in immune evasion and suppression in HNSCC and that targeting NTMT1 may enhance the effectiveness of immunotherapies in these cancers.

Correlation and enrichment analyses

To gain further insight into the molecular mechanisms underlying NTMT1’s role in tumorigenesis, we aimed to identify proteins that interact with NTMT1 and genes that are associated with NTMT1 expression for pathway enrichment analyses. We utilized the BioGRID database to uncover novel proteins and pathways associated with NTMT1. By constructing a network of 68 NTMT1-associated proteins through BioGRID (Fig. 8A), we gained valuable insights into the molecular interactions and pathways involving NTMT1. We also utilized the STRING tool to identify 50 NTMT1 binding proteins supported by experimental evidence, as shown in Fig. 8B. Obtain the top 100 genes (Table S1) associated with NTMT1 through GEPIA2, and perform an intersection analysis with genes related to HNSCC, resulting in 28 genes. The correlation between HNSCC and NTMT1 expression of these genes is shown in Fig. 8C. This information can help elucidate the function of NTMT1 in biological processes and diseases and potentially identify new therapeutic targets for cancer treatment.

Figure 8 NTMT1-related gene enrichment analysis.

(A) NTMT1-related genes were obtained from the BioGRID web, and 68 proteins were displayed. (B) Network of 50 NTMT1-related protein form STRING. (C) The heat map of NTMT1 and its co-expressed mRNA in HNSCC. ***p < 0.001. (D) Volcano plot of differentially expressed genes (DEGs) between high and low NTMT1 expression. (E) The mountain plot of GSEA analysis for DEGs. (F) Circle plot showing the GO and KEGG enrichment analysis results for DEGs between high and low NTMT1 expression groups.

Additionally, we conducted a correlation analysis between NTMT1 and all other mRNAs using TCGA data to explore the function and pathways influenced by NTMT1. We analyzed the differentially expressed genes between the high and low expression groups of NTMT1 in HNSCC, and selected genes with |logFC| ≥ 1 for further analysis (Fig. 8D). Gene Ontology (GO) and Kyoto Encyclopedia of Genes and Genomes (KEGG) analyses revealed that these genes are associated with various immune-related processes (Fig. 8F), such as immune response-activating cell surface receptor signaling pathway, phagocytosis, activation of the immune response, external side of the plasma membrane, intermediate filament cytoskeleton, T cell receptor complex, antigen binding, calcium-dependent phospholipase A2 activity, C-C chemokine receptor activity, cytokine-cytokine receptor interaction, estrogen signaling pathway, and vascular smooth muscle contraction.

Further analysis using GSEA revealed that the genes correlated with NTMT1 expression are functionally related to various pathways, such as FCERI_MEDIATED_MAPK_ACTIVATION, FCERI_MEDIATED_NF_KB_ACTIVATION, ROLE_OF_PHOSPHOLIPIDS_IN_PHAGOCYTOSIS, ADAPTIVE_IMMUNE_SYSTEM, and INNATE_IMMUNE_SYSTEM (Fig. 8E). These findings suggest that NTMT1 may modulate the immune response and phagocytosis in HNSCC and may also regulate various signaling pathways. The identified pathways could offer potential targets for therapeutic intervention in HNSCC.

The expression pattern of NTMT1 at single-cell levels

ScRNA-seq provides an unprecedented opportunity to explore the functional heterogeneity of cancer cells. NTMT1 expression is dysregulated in various types of cancer, and its overexpression or underexpression has been associated with different types of cellular functions. The results suggest that NTMT1 expression is positively associated with various cellular functions, such as cell cycle, DNA damage, DNA repair, invasion metastasis, and apoptosis in ALL, AML, CML, CRC, BRCA, AST, GBM, Glioma, HGG, ODG, HNSCC, LUAD, NSCLC, OV, and MEL (Fig. 9A). The expression of NTMT1 is positively correlated with cell proliferation and DNA repair in HNSCC (Figs. 9B and 9C), which is consistent with previous literature reports. NTMT1 expression is upregulated in several cancers, including HNSCC, and its overexpression has been associated with increased tumor growth and invasion. Therefore, the positive correlation between NTMT1 expression and cell proliferation in HNSCC may indicate that NTMT1 could be a potential therapeutic target for this cancer type.

Figure 9 The expression levels of NTMT1 at single-cell levels.

(A) The relationship between NTMT1 expression and different functional states in tumors by the CancerSEA tool. (B and C) The relationship between NTMT1 expression and proliferation and DNA repair in HNSCC. *p < 0.05, **p < 0.01. (D) NTMT1 expression profiles were shown at single cells from HNSCC by T-SNE diagram.

However, it is primarily negatively associated with the cellular functional states of RCC, RB, and UM. In particular, NTMT1 expression is significantly and negatively correlated with 14 functional states in UM (Fig. 9A), suggesting that it may act as a potential tumor suppressor in UM. Further studies are needed to elucidate the underlying mechanisms of NTMT1 in UM. Additionally, the T-SNE plot displays the single-cell level expression profiles of NTMT1 in HNSCC (Fig. 9D), indicating that there are differences in NTMT1 expression among these cancer types at the single-cell level. However, the exact nature of these differences is not specified in the text provided.

Overall, these findings suggest that NTMT1 expression may play a complex and context-dependent role in cancer biology, potentially affecting multiple cellular processes that contribute to the development and progression of cancer. However, further research is needed to fully elucidate the functional significance of NTMT1 expression in cancer and its potential as a therapeutic target.

NTMT1 knockdown promoted tumor cell apoptosis

To further explore the role of NTMT1 in HNSCC, we knocked down NTMT1 expression by siRNA (Fig. 10A). The results showed that NTMT1 knockdown significantly inhibited the proliferation of HNSCC cell SNU1076 (Fig. 10B). Since knockdown of siNTMT1-2 could more effectively inhibit the expression of NTMT1 and the proliferation of tumor cells, siNTMT1-2 was used for further study in subsequent experiments. Furthermore, the immunofluorescence assay showed that the expression of γ-H2AX was significantly increased after NTMT1 knockdown (Fig. 10C) indicating that the cells had significant DNA damage. Flow cytometry showed that NTMT1 knockdown promoted tumor cell apoptosis (Figs. 10D and 10E). These results suggest that knockdown of NTMT1 may promote cell apoptosis by inducing DNA damage response in HNSCC. This is consistent with the single-cell analysis results. However, further specific mechanisms need to be explored.

Figure 10 Effect of NTMT1 on HNSCC tumor cell.

(A) SNU1076 cell was transfected with siNTMT1 for 72 h, and the mRNA level of NTMT1 was evaluated by RT-qPCR. n = 3, data represent mean ± SEM. ns, no significant, ***p < 0.001. (B) The proliferation of SNU1076 was examined by CCK-8 assay. n = 3, data represent mean ± SEM. ns, no significant, ***p < 0.001. The experiment was repeated three times. (C) IF images of γ-H2AX in SNU1076 cell transfected with siNTMT1 or siNC for 72 h. Scale bar = 10 μm. Nuclei appear blue (DAPI). (D and E) Flow representative (D) and statistical analysis (E) of apoptosis in SNU1076 cells treated for 72 h. Data represent the mean ± SEM. **p < 0.01. siNC was used as control group, and p < 0.05 was considered statistically significant.

Discussion

NTMT1 is a newly discovered protein that has the potential to play a crucial role in various cellular biological processes (Webb & Lipson, 2010). NTMT1 is an alpha-N-methyltransferase that methylates RCC1 and retinoblastoma protein (Tooley & Petkowski, 2010). Nonetheless, even with ongoing research, its particular function remains unclear. Recent research has also indicated that NTMT1 may participate in the initiation and progression of tumors. Its function as a tumor suppressor gene or an oncogene depends on the specific tissue and the pathway that drives cancer development (Deng & Su, 2018). In this study, we explored the potential role of NTMT1 in cancer using pan-cancer bioinformatic analysis.

Our results revealed that both mRNA and protein levels of NTMT1 were markedly increased in various types of tumors, and high expression of NTMT1 was closely correlated with poor prognosis in ACC, HNSCC, AML, READ, and UVM. The previous literature has reported a correlation between NTMT1 and the development of AML, LIHC, and CESC (Yankova & Blackaby, 2021; Zhang & Song, 2021; Pan & Lin, 2021; Campeanu & Jiang, 2021; Tooley & Catlin, 2023). Our exploratory findings also suggest that alterations in the NTMT1 gene, including mutations and deep deletions, can be observed in various types of tumors. Despite a low mutation rate of only 2.4%, a high frequency of mutations is associated with poor prognosis. Additionally, significant differences in the methylation levels of NTMT1 were observed between various tumor tissues and normal tissues, indicating that NTMT1 may play a crucial regulatory role in tumorigenesis. The specific role and potential mechanisms of NTMT1 in human cancer are not yet clear and require further exploration. For instance, the active inhibitor will serve as a valuable tools to examine the physiological and pharmacological functions of NTMT1 catalytic activity. It has been suggested that NTMT1 may be counterregulated by its family members METTL11B and METTL13 (Parker & Schaner Tooley, 2023). In addition, Meghan’s group confirmed the misregulation of CYP and MUP mRNA and protein levels in Nrmt1-/- livers and verified that NRMT1 can methylate ZHX2 in vitro (Conner & Parker, 2022).

Considering the significant breakthroughs in immunotherapy in cancer treatment in recent years, the correlation between NTMT1 and tumor immune infiltration is crucial (Galluzzi & Chan, 2018). Our study results suggest that NTMT1 expression is positively correlated with various immune-related genes in some tumors, including BLCA, PCPG, SARC, and LGG. This suggests that NTMT1 expression may serve as a useful biomarker for identifying patients who could benefit from immunotherapy. Furthermore, our findings indicate that NTMT1 expression does not affect the prognosis of these tumor types, highlighting the complexity of NTMT1’s role in regulating tumorigenesis. However, in most cancers, including HNSCC, NTMT1 is negatively correlated with chemokine receptors. In previous research, Li’s group found the complexity of the immune microenvironment in HNSCC (Li & Liu, 2022). Through an in-depth analysis of NTMT1’s immune-relatedness in HNSCC, our results show that NTMT1 expression is significantly and negatively correlated with various immune cell-related molecules and immune scores. This suggests that NTMT1 may be involved in immune escape or immune heterogeneity within the microenvironment of HNSCC. There is research that has utilized the targeted protein degradation strategy called proteolysis-targeting chimeras (PROTAC) to synthesize a drug degrader 1 that can degrade NTMT1 expression (Zhou & Wu, 2022). This drug degrader can induce the overexpression of calreticulin (CALR), an immunogenic cell death (ICD) signaling protein. CALR is associated with CTL infiltration in three types of tumors (colorectal, breast, and ovarian), suggesting that the loss of CALR expression may negatively impact immune surveillance and reduce patient survival rates (Stoll & Iribarren, 2016). Additionally, CALR is clinically associated with high survival rates in colorectal cancer patients (Arai & Xiao, 2020), indicating that a combination of NTMT1 protein degradation drugs and immune checkpoint therapy may improve the efficacy of tumor treatment. The results of functional enrichment analysis indicate that genes co-expressed with NTMT1 are mainly involved in biological processes such as immune response, and activation of NF-κB and MAPK signaling pathways, which are closely related to the occurrence and development of cancer. This suggests that NTMT1 may play an important regulatory role in the tumor immune microenvironment, but further experimental exploration is necessary to confirm this hypothesis.

The infiltrative growth and continuous proliferation of malignant tumors make them difficult to cure, and exploring the potential mechanisms of their malignancy is imperative. Single-cell functional analysis of tumors reveals that NTMT1 is associated with various biological behaviors of tumor cells, such as stemness, invasion, proliferation, metastasis, apoptosis, and EMT, which promote tumor development. However, NTMT1 is negatively correlated with DNA repair. A small molecule inhibitor targeting METTL3 has been reported to have a significant therapeutic effect on AML (Yankova & Blackaby, 2021). Considering the significant role of NTMT1 in AML development, the development of NTMT1 inhibitors may provide valuable insights for the treatment of AML. We also found that NTMT1 is associated with HNSCC stemness, cell proliferation, DNA repair, and cell cycle, but the specific role of NTMT1 in HNSCC has not been reported. In this study, we knocked down NTMT1 gene expression and found that it significantly inhibited tumor cell proliferation, induced DNA damage, and ultimately promoted tumor cell apoptosis.

Fortunately, the development of NTMT1 inhibitors is continuously being explored and has made significant progress, as evidenced by previous studies (Mackie & Chen, 2020; Dong & Deng, 2022; Chen & Meng, 2021; Chen & Dong, 2019, 2021). This study utilized comprehensive bioinformatics analysis techniques to investigate the expression level, clinical prognostic value, immune relevance, and functional enrichment of NTMT1 in pan-cancer. In addition, we conducted preliminary in vitro experiments to investigate the role of NTMT1 in HNSCC development. However, further exploration is needed to investigate the underlying mechanisms in more depth.

Conclusions

In conclusion, this study has explored the important role of NTMT1 in cancer. Our findings provide a foundation for further research into the specific mechanisms of NTMT1 in different tumor development and treatment, suggesting its potential as an important target for cancer treatment and immunotherapy. However, larger and more comprehensive datasets, as well as diverse experimental validations, are needed to support our conclusions. Further verification of the biological functions and regulatory mechanisms of NTMT1 is necessary to better understand its role in cancer and can provide new ideas and strategies for cancer treatment and prognosis evaluation.

Supplemental Information

Supplemental Information 1 TMT1 protein expression in NTMT1 protein expression in various tumors.

(A) NTMT1 protein levels in variety of tumors from UALCAN. (B) Protein expression of NTMT1 across pan cancer subtype. (C) Protein expression of NTMT1 across pan cancer subtype2.

Click here for additional data file.

Supplemental Information 2 Prognostic values of NTMT1 expression in pan-cancer from Kaplan-Meier plotter.

(A) Effects of NTMT1 expression on OS in BLCA, HNSCC and STAD, respectively. (B) Effects of NTMT1 expression on RFS in LIHC, LUSC and STAD, respectively.

Click here for additional data file.

Supplemental Information 3 Promoter methylation levels of NTMT1 in pan-cancer.

The methylation level of the NTMT1 promoter region in BRCA, CHOL, PAAD, LUAD, KIRP, COAD, READ, SARC, and THCA from UALCAN.

Click here for additional data file.

Supplemental Information 4 Correlation of NTMT1 and immuno‑related genes in HNSCC.

(A) The relationship of NTMT1 expression and infiltration of immune cells in HNSCC. (B) The relationship of NTMT1 expression and infiltration of stromal score in HNSCC. (C–G) The relationship of NTMT1 expression and chemokines (C), immunostimulators (D), chemokines receptors (E), MHC molecules (F), and immunoinhibitors (G) in HNSCC. ns, p＞0.05, *p＜0.05, **p＜0.01, ***p＜0.001.

Click here for additional data file.

Supplemental Information 5 The top 100 genes associated with NTMT1 through GEPIA2.

Click here for additional data file.

Supplemental Information 6 Raw data for Figure 10.

-

Click here for additional data file.

Additional Information and Declarations

Competing Interests

Author Contributions

Data Availability

The authors declare that they have no competing interests.

Lifan Tan conceived and designed the experiments, performed the experiments, analyzed the data, prepared figures and/or tables, authored or reviewed drafts of the article, and approved the final draft.

Wensong Li analyzed the data, prepared figures and/or tables, and approved the final draft.

Qin Su conceived and designed the experiments, authored or reviewed drafts of the article, and approved the final draft.

The following information was supplied regarding data availability:

The raw data of Fig. 10 is available in the Supplemental File.

The mRNA expression profiles and clinical outcome data are available at The Cancer Genome Atlas (TCGA) (https://portal.gdc.cancer.gov/) and the University of California Santa Cruz (UCSC) Xena (https://xenabrowser.net/datapages/).

The expression levels of NTMT1 protein are available at Human Protein Atlas (HPA) database (http://www.proteinatlas.org) and UCLCAN databases (https://ualcan.path.uab.edu/index.html). The single-cell sequencing data are available at CancerSEA (http://biocc.hrbmu.edu.cn/CancerSEA/home.jsp).

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
