# Peer review of "The comprehensive analysis of the prognostic and functional role of N-terminal methyltransferases 1 in pan-cancer"

_PeerJ, doi:10.7717/peerj.16263_

## Round 0.1 · original submission · Minor Revisions

Please address the critiques of all reviewers and amend the manuscript accordingly.

Reviewer 1 ·

Basic reporting

The paper is written in a professional English language. All necessary references, background, figures, tables and raw data are provided. There may be a typo in the beginning of line 197.

Experimental design

In line 283-284, the authors stated that "the AUC value of 0.878 for HNSCC suggests that NTMT1 expression may not be as reliable of a diagnostic biomarker for HNSCC compared to the other types of cancer". Usually AUC=0.878 is considered as excellent. Please further justify this argument.

Validity of the findings

No comment.

Additional comments

This is an informative and complete work. I would recommend acceptance with all above minor comments addressed.

Reviewer 2 ·

Basic reporting

This manuscript investigates the potential role of NTMT1 in cancer development and progression. Authors performed several bioinformatics analysis using various cancer datasets, the study reveals that NTMT1 expression is elevated in certain cancer types and is directly linked with poor prognosis. The authors also identify associations between NTMT1 expression and immune-related genes, suggesting its relevance in immunotherapy strategies.
The introduction has limitations and could be improved. Please look at the following suggestions:

1. Include how NTMT1's methyltransferase activity might influence key cancer hallmarks, such as proliferation, apoptosis, DNA repair, and metastasis. Authors could include examples from previous research articles.

2. Introduce the concept of targeting NTMT1 for cancer therapy, citing recent studies that explore the potential of NTMT1 inhibition as a therapeutic strategy and discussing the rationale behind this approach.

3. Address the gaps in understanding NTMT1's functions, such as its role in specific cancer subtypes, the underlying molecular mechanisms, and its interactions with other cellular components.

Experimental design

The authors utilized various publicly available databases such as TCGA and GTEx for gene expression analysis and clinical information. They performed statistical analyses to identify significant differences in NTMT1 expression across different cancer types and correlated it with clinical outcomes. Advanced bioinformatics tools were employed for data analysis, including differential expression analysis, survival analysis, and correlation studies. The authors conducted single-cell analysis to explore NTMT1's impact on various tumor cell behaviors. In vitro experiments involved siRNA knockdown of NTMT1 to assess its effects on cell proliferation and apoptosis.
Authors should include specifics about data preprocessing, software tools, and experimental controls to ensure reproducibility of the results. Including potential biases and limitations would strengthen the methodology.

Validity of the findings

The authors found significant overexpression of NTMT1 across various cancers and highlighted its correlation with poor prognosis in specific types. NTMT1 mutations, deletions, and methylation alterations were explored, suggesting its involvement in tumorigenesis. Immune-related analyses revealed potential implications for immunotherapy. Single-cell analysis indicated NTMT1's diverse roles in different tumor behaviors. The authors also investigated the role of NTMT1 in cancer biology, its potential as both a tumor suppressor and an oncogene, and its implications for immunotherapy. However, explanations of mechanisms and examples of certain processes with references could be added for more clarity.

·

Basic reporting

The author investigated the role of the N-terminal methyltransferases 1 (NTMT1) gene in a panel of cancer cell lines through both bioinformatics techniques and proliferation inhibition assay on the model cell line, HNSCC, the head and neck squamous cells. The results indicated that NTMT1 was involved in cell proliferation progression and inhibition of cancer immunity in HNSCC and some other cell lines. The article has a good layout in general and the study background was well illustrated. The figure quality is acceptable as well.

Experimental design

The data sets analyzed was convincing enough to support the author’s hypothesis. An array of bioinformatics techniques including various databases showed that NTMT1 has a higher expression level in some of the cancer cell lines. Besides, using HNSCC cell line as a model, the author applied genome sequencing as well as immunoassays to illustrate the importance role of NTMT1 in inhibiting cell apoptosis as well as DNA damage.

Validity of the findings

The importance of NTMT1 gene in cancer study was proved by the results presented. Although the pan-cancer system is complicated, the author has paved a solid road towards the future study in the field.

Additional comments

In general the paper is well written, and presents a convincing argument for its publication for PeerJ Life and Environment journal. However, I would suggest that the authors conduct a thorough review of the document and correct for the typographical and/or grammatical errors.

Examples are listed:

- Figure 10C was not bold
- Line 443 and 444 “The results suggest that NTMT1 may promote cell apoptosis by inducing DNA damage response in HNSCC.” was not consistent with the experiment conclusion. The knockdown of NTMT1 should be the subject.
- In figure 10 description, qRT-PCR should be RT-qPCR
- Confusing and inconsistent information - for example, in the “Method” part, cells were treated with siRNA for 48 hours but in the “Figure 10 description” the treatment was 72 hours.
- Since two siRNA was used for the knockdown experiment, the difference of siNTMT1-1 vs siNTMT1-2 should be mentioned in the article illustrating that siNTMT1-1 was not effective as siNTMT1-2.

---

## Round 0.2 · accepted · Accept

All reviewers were satisfied by the revision and I am happy to accept your amended manuscript.

Reviewer 1 ·

Basic reporting

no comment

Experimental design

no comment

Validity of the findings

no comment

Additional comments

The authors responded to all my previous comments. The revision version is drastically improved. I would suggest accept.

Reviewer 2 ·

Basic reporting

Authors investigated the role of NTMT1 in development and progression of cancer. Several bioinformatics analysis are performed using cancer datasets suggesting the elevation of NTMT1 expression in various cancer types and is linked to poor prognosis.

Experimental design

authors performed various statistical analyses using publicly available datasets to study the NTMT1 expression in various cancer types and their correlation with the clinical outcomes.

Validity of the findings

Authors validated over-expression of NTMT1 across various cancers. Authors studied the mutations, deletions, methylation patterns to determine the involvement of NTMT1 in cancer progression. Authors also explored the role of NTMT1 as both tumor suppressor and and oncogene. Immune related analysis was also performed to determine the implication in immunotherapy.

Additional comments

Authors response to the comments are satisfactory. The manuscript can be published as is.

·

Basic reporting

The author investigated the role of the N-terminal methyltransferases 1 (NTMT1) gene in a panel of cancer cell lines through both bioinformatics techniques and proliferation inhibition assay on the model cell line, HNSCC, the head and neck squamous cells. The results indicated that NTMT1 was involved in cell proliferation progression and inhibition of cancer immunity in HNSCC and some other cell lines.

The article has a good layout in general and the study background was well illustrated. The figure quality is acceptable as well.

Experimental design

The data sets analyzed was convincing enough to support the author’s hypothesis. An array of bioinformatics techniques including various databases showed that NTMT1 has a higher expression level in some of the cancer cell lines. Besides, using HNSCC cell line as a model, the author applied genome sequencing as well as immunoassays to illustrate the importance role of NTMT1 in inhibiting cell apoptosis as well as DNA damage.

Validity of the findings

The importance of NTMT1 gene in cancer study was proved by the results presented. Although the pan-cancer system is complicated, the author has paved a solid road towards the future study in the field.

Additional comments

The author has addressed all of the comments mentioned in the previous review. The content is more coherent with a smooth logic.